# Metric clusters in evolutionary games on scale-free networks

Kaj-Kolja Kleineberg [1]

The evolution of cooperation in social dilemmas in structured populations has been studied extensively in recent years. Whereas many theoretical studies have found that a heterogeneous network of contacts favors cooperation, the impact of spatial effects in scale-free networks is still not well understood. In addition to being heterogeneous, real contact networks exhibit a high mean local clustering coefficient, which implies the existence of an underlying metric space. Here we show that evolutionary dynamics in scale-free networks self-organize into spatial patterns in the underlying metric space. The resulting metric clusters of cooperators are able to survive in social dilemmas as their spatial organization shields them from surrounding defectors, similar to spatial selection in Euclidean space. We show that under certain conditions these metric clusters are more efficient than the most connected nodes at sustaining cooperation and that heterogeneity does not always favor—but can even hinder—cooperation in social dilemmas.

[1] Computational Social Science, ETH Zurich, Clausiusstrasse 50, CH-8092 Zurich, Switzerland. Correspondence and requests for materials should be addressed to K.-K. (email: kkleineberg@ethz.ch)

Cooperation among humans has been found to be quite common in social dilemmas[1, 2], and plays a major role in the emergence of complex modern societies[3, 4]. Therefore, understanding the underlying mechanisms that can give rise to and sustain cooperation from an evolutionary perspective is key to complementing Darwin's theory of evolution[5–9].

In reality, populations are structured, which means that the topology of strategic interactions is given by a network of contacts. In structured populations, individuals interact repeatedly with the same individuals. Thus, as a consequence, cooperators can survive in social dilemmas by forming network clusters. This mechanism is referred to as network reciprocity[2, 10]. In the well-studied case of lattice topologies, the resulting network clusters unfold in Euclidean space[11–13] (spatial selection[14]). Realistic networks of contacts are heterogeneous rather than lattices and often scale-free, which means that their degree distribution follows a power-law with exponent $\gamma \in (2, 3)$, where a lower value of $\gamma$ means more heterogeneous networks. Heterogeneity has been shown to favor cooperation[15–17], and cooperating nodes form a connected (or network) cluster[18]. However, the geometric organization of these connected clusters—similarly to spatial selection in Euclidean space—remains elusive.

Real complex networks, in addition to being heterogeneous, exhibit a high mean local clustering coefficient[19, 20] (this means that the network contains a high number of closed triangles). This is particularly important because a high clustering coefficient implies the existence of an underlying metric space[21].

We show that evolutionary dynamics on scale-free, highly clustered networks lead to the formation of patterns in the underlying metric space, similar to the aforementioned spatial selection in Euclidean space. Using two empirical networks, the IPv6 Internet topology and the arXiv collaboration network, as well as synthetic networks, we show that spatial patterns play an important role in the evolution of cooperation. In fact, under certain conditions metric clusters can even be more effective at sustaining cooperation than the most connected nodes (hubs). As

a consequence, heterogeneity does not always favor—but can even hinder—the evolution of cooperation in social dilemmas.

## Results

**Latent geometry of scale-free networks.** Real contact networks are usually heterogeneous, and often scale-free, as well as highly clustered[20] (we refer to a high mean local clustering coefficient, i.e. a large number of closed triangles[19]). The effect of scale-free topologies has attracted a lot of attention, and many theoretical studies have found that heterogeneous networks of contacts favor cooperation in social dilemmas[15–18], although this behavior has not been confirmed in recent experiments with human players[22]. Importantly, the high local clustering coefficients found in real contact networks have been proven to imply the existence of a metric space underlying the observed topology[21]. This means that the nodes of a given real complex network can be mapped to coordinates in this metric space such that the probability that pairs of nodes will be connected in the observed topology depends only on their distance in the metric space. Specifically, heterogeneous networks can be embedded into hyperbolic space[23–25]. In this representation, each node has a radial and angular coordinate. The radial coordinate abstracts the popularity, and hence the degree of the node, such that hubs are placed closer to the center of the disc (Fig. 1a). The angular coordinate abstracts a similarity space, such that the angular distance is a measure of the similarity between two nodes, whereby nodes tend to connect to more similar nodes. In Fig. 1a, b we show an illustration of the hyperbolic metric structure underlying two different networks (see Methods section for further details). In the following, we show that evolutionary dynamics trigger the formation of stable spatial clusters in the angular dimension on the underlying hyperbolic space.

**Evolutionary dynamics and the emergence of metric clusters.** Let us first consider the prisoner's dilemma game, in particular $T = 1.2$ and $S = -0.2$ (see Eq. (1) in Methods) on synthetic contact

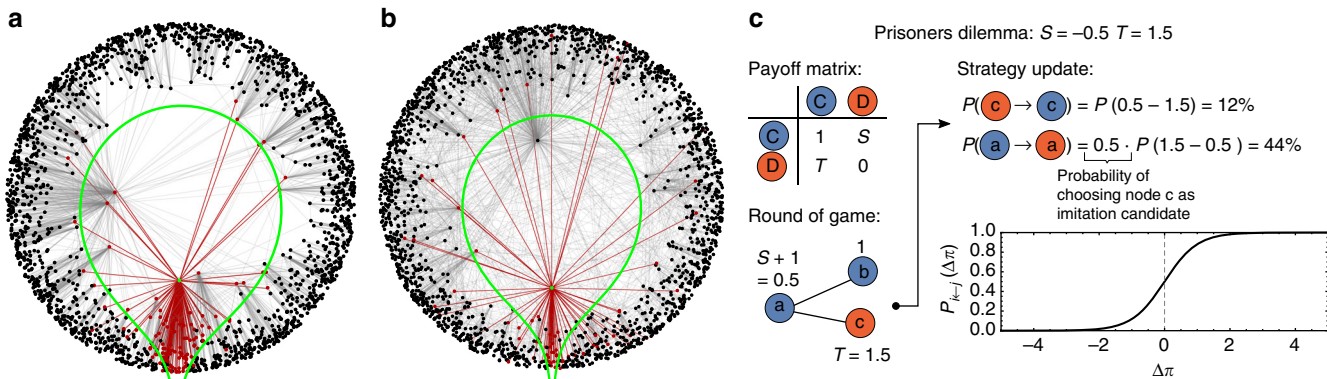

**Fig. 1** Evolutionary dynamics on scale-free networks embedded into hyperbolic space. **a** Illustration of the hyperbolic spatial structure (Poincaré Disc) underlying a synthetic network generated by the model described in Methods. The network shown here has $N = 2000$ nodes, a power-law exponent $\gamma = 2.6$, mean degree $\langle k \rangle \approx 6$, and mean local clustering coefficient $\bar{c} = 0.6$ (temperature $\bar{T} = 0.3$ as in Eq. (6), see Methods section for details). Hubs, i.e. high-degree nodes, are placed closer toward the center of the disc (lower radial coordinate). The angular space represents the similarity between nodes, such that nodes tend to connect to other nodes close to them in this space. The green line shows the hyperbolic disc of radius $R$ (Eq. (5)) around the green node. We highlight the neighbors of the green node in red. For a high $\bar{c}$ (i.e., low $\bar{T}$), as shown here, the green node is highly likely to connect to other nodes within the disc (green line), and very unlikely to connect to nodes outside of it (the further apart, the less probable). The high mean local clustering coefficient is then a consequence of the triangle inequality in the metric space. **b** Synthetic network generated with the same model but with mean local clustering coefficient $\bar{c} = 0.25$ (temperature $\bar{T} = 0.7$). We again show the hyperbolic disc of radius $R$ (Eq. (5)) around the green node and highlight its neighbors in red. Note that due to the higher temperature more long-range connections are formed, i.e., the green node connects to more nodes outside of the disc as compared to **a**, and does not connect to some node inside of the disc. This effect reduces the mean local clustering coefficient as it induces randomness in the link formation process. **c** Illustration of evolutionary game dynamics. In structured populations, individuals play with their neighbors in a network. In each game, they generate a payoff given by the payoff matrix (Eq. (1)). After each round, they choose a random neighbor and imitate her strategy with a probability (Fermi–Dirac distribution) that depends on the difference between their payoffs

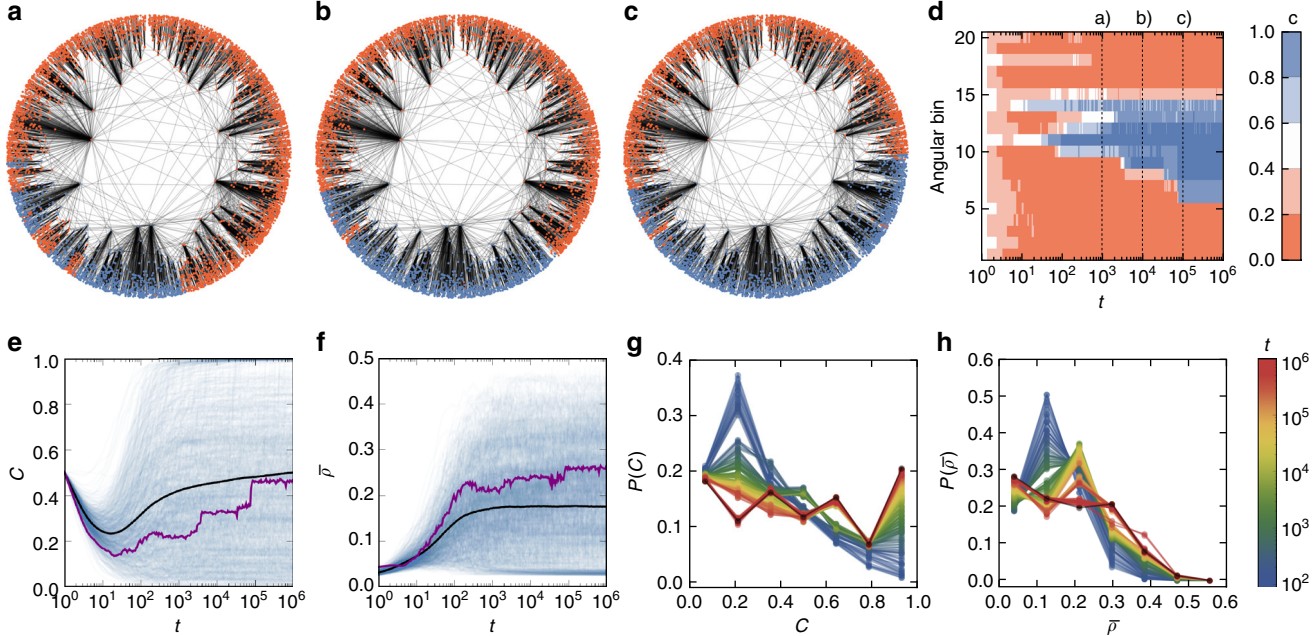

**Fig. 2** Spatial patterns in evolutionary games. **a–c** Evolution of the system (see also Supplementary Movie 1) for a single realization of the prisoner's dilemma ($T = 1.2$ and $S = -0.2$). Here we have generated a synthetic network with $N = 5000$ nodes, power law exponent $\gamma = 2.8$, and mean local clustering $\bar{c} \approx 0.5$. Cooperators are marked in blue, whereas red denotes defectors. The system was started with randomly selected cooperators ($c(0) = 0.5$). **a** shows the state of the system after $10^3$, **b** after $10^4$, and **c** after $10^5$ generations. **d** Density of cooperators (color coded) in different angular bins (shown on the y axis) as a function of time (shown on the x axis). Here we have divided the angular space $\theta \in [0, 2\pi)$ into 20 equidistant bins. **e–h** Results for the same synthetic networks as before but with $N = 2 \times 10^4$ nodes. **e** Evolution of the density of cooperation $C$ for $10^3$ independent realizations of the system (blue lines) and their average (black line). The purple line corresponds to the realization shown in **a–c**. **f** Evolution of the KS-statistics $\bar{\rho}$ for $10^3$ independent realizations of the system (blue lines) and their average (black line). The purple line corresponds to the realization shown in **a–c**. **g** Evolution of the distribution of cooperation $C$ observed in the system. Each color represents the histogram of cooperation among different realizations of the system at time $t$, where the color denotes the time (see legend in plot **h**). The 7 points hence represent 7 bins. The x-axis denotes the cooperation density $C$ and the y-axis shows the probability $P(C)$ that in one realization of the system the density of cooperation at time $t$ is $C$. **h** Evolution of the distribution of $\bar{\rho}$ observed in the system (colors denote time)

networks generated with the model described in Methods. This model generates realistic topologies based on underlying hyperbolic metric spaces, similar to Fig. 1a. We simulate the evolutionary game dynamics (see Methods and Fig. 1c) and find that the system tends to self-organize into a state in which groups that are mainly cooperative are clearly separate from groups populated mainly by defectors (see Fig. 2a–c and Supplementary Movie 1)[26, 27]. Similarly to the case of lattice topologies and spatial selection in Euclidean space, we observe the formation of clusters of cooperators in the angular dimension of the underlying hyperbolic space. In Fig. 2d, we show the evolution of the density of cooperators in different bins of the angular coordinate $\theta$. We observe that initially cooperation decreases (see purple line in Fig. 2e) while, at the same time, the remaining cooperators become concentrated in clusters in the angular space (Fig. 2a). Cooperation then increases again as it spreads in the vicinity of the clusters (Fig. 2b) until the system reaches a stationary state with fluctuations only at the borders of the clusters (Fig. 2c and Supplementary Movie 1).

We quantify the degree to which cooperators and defectors cluster in the angular space using the Kolmogorov–Smirnov (KS) statistic[28], which measures the difference between two one-dimensional distributions. The KS statistic is defined as the maximum absolute difference between the values of two cumulative distributions. In particular, we define the KS statistic of the distribution of cooperation density in different angular bins, and the uniform distribution at time $t$ as $\bar{\rho}(t)$ (see Methods for details). A higher value of $\bar{\rho}(t)$ thus denotes more pronounced clustering of cooperators and defectors, respectively. In Fig. 2f, we

show the evolution of $\bar{\rho}(t)$ for $10^3$ different realizations (blue lines) and their mean (black line). On average, $\bar{\rho}(t)$ increases initially and approaches a constant value after $\sim 10^2$–$10^3$ generations. Among different realizations, $\bar{\rho}(t)$ varies significantly and we study the evolution of its distribution in Fig. 2h. We find that at relatively low times, the distribution shows a peak at $\bar{\rho} \approx 0.2$, which then declines. Eventually (black line), there is a high proportion of realizations with $\bar{\rho} \approx 0$, which must be the case if the system approaches a state with nearly full cooperation or defection. It can also be observed from the evolution of the distribution of cooperation (Fig. 2g) that the probability of high cooperation $C \approx 1$ increases over time. In combination, these observations indicate that the evolutionary path toward full cooperation includes a phase of significant clustering of cooperators. The stationary distribution of $\bar{\rho}$ (colored lines converge to the black line in Fig. 2h) shows that apart from the aforementioned realizations, the values of $\bar{\rho}$ are distributed around the mean of $\bar{\rho} \approx 0.2$.

To conclude, evolutionary dynamics on scale-free networks lead to the formation of stable spatial patterns, which can be observed as metric clusters in the angular dimension of underlying hyperbolic metric spaces. This behavior is similar to spatial selection in lattice topologies, where cooperators form spatial clusters in Euclidean space.

Let us illustrate the difference between the discovered metric clusters and connected clusters. A connected cluster, i.e. a subgraph in which each pair of nodes is connected using only paths among the nodes of the subgraph[19], is shown in Fig. 3a (blue nodes). This connected cluster was generated following the

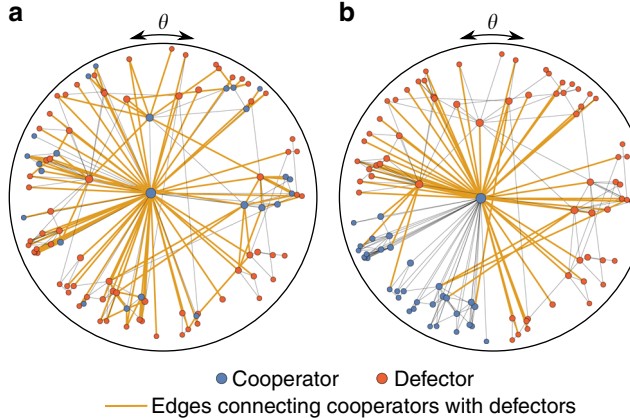

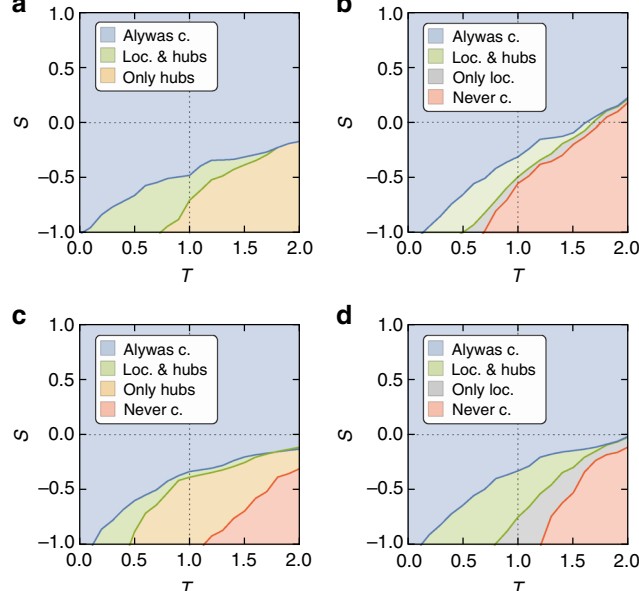

**Fig. 3** Difference between connected and metric clusters. Illustration of a connected cluster and a metric cluster using a small network generated with the model described in Methods. **a** Cooperators (blue nodes) form a connected cluster, which has been assigned following the procedure described in Methods (here we assigned $N/4$ nodes into the connected cluster). **b** The same network as in **a**, but now cooperators form a metric cluster, i.e., their similarity (angular) coordinates $\theta$ lie in the interval $[\pi, 3/2\pi)$

**Fig. 4** Game parameter phasespace for different topologies. We show the final density of cooperators (after $2 \times 10^5$ update steps) averaged over 50 realizations of the system, as a function of the game parameters $T$ and $S$ from Eq. (1). Colors denote regions in the parameter space where final cooperation exceeds the threshold value of 0.3. This is always the case in the blue area, only if started with either cooperative hubs or a metric cluster in the green region, only if started with cooperative hubs in the yellow area, only if started with metric clusters in the gray area, and for none of the considered initial conditions in the red region. **a** Results for the IPv6 Internet topology. **b** Results for the arXiv collaboration network. **c** Synthetic network with $N = 2 \times 10^4$ nodes, power-law exponent $\gamma = 2.4$, mean degree $\langle k \rangle \approx 6$, and clustering $\bar{c} = 0.5$. **d** The same as before but for networks with power-law exponent $\gamma = 2.9$ and clustering $\bar{c} = 0.6$

procedure described in Methods. The similarity (angular) coordinates $\theta$ of the nodes belonging to this connected cluster are uniformly distributed. A metric cluster is illustrated in Fig. 3b. Contrarily to the previous case, the similarity (angular) coordinates $\theta$ of nodes belonging to the metric cluster by definition lie in a certain interval, here chosen to be $[\pi, 3/2\pi)$ (see also Supplementary Fig. 3). Note that the metric cluster in Fig. 3b is also a connected cluster, but the connected cluster in Fig. 3a is not a metric cluster, because the similarity coordinates of its nodes are not constraint to a certain interval in the angular space. This example also allows us to understand why, as we will show, the metric cluster is more effective in shielding cooperators from defectors (akin to spatial selection) than the connected cluster. This is due to the different abundance of intercluster links, i.e., links that connect a cooperator and a defector, which we highlight in Fig. 3. There are 125 of such links in the example of the connected cluster and only 53 for the metric cluster, where intercluster links mainly occur at the border of the metric cluster (this effect is more pronounced for larger networks). We analyze the abundance of intercluster links in detail later.

**Metric clusters can be more effective than hubs**. Let us now consider two empirical networks, the Internet Ipv6 topology, which has $N = 5162$ nodes, a degree distribution with power-law exponent $\gamma = 2.1$, average degree $\bar{k} = 5.2$, and a mean local clustering coefficient of $\bar{c} = 0.22$ and the arXiv collaboration network, which has 1905 nodes, mean degree $\langle k \rangle = 4.6$, mean local clustering coefficient $\bar{c} = 0.66$, and a power-law degree exponent $\gamma = 3.9$. To address the question of whether spatial clusters or the hubs of a network are more efficient at sustaining cooperation, we use the initial conditions as a proxy for possible control mechanisms[26, 29–31]. Specifically, we distribute the initial cooperators (always $c(0) = 0.5$) in the system as follows: first, we randomly assign 50% of the nodes as cooperators; second, we assign the same number of cooperators preferentially to the hubs of the system, i.e., we select nodes proportional to their degree; third, we assign the same number of cooperators into a metric cluster in the similarity space (see Methods). The first strategy serves as a null model, the second mimics the potential of the hubs to drive the system toward cooperation, while the third

strategy serves as a proxy for the ability of metric clusters of cooperators to survive.

Figure 4a shows the result for the Internet IPv6 topology, where we show the regions in the $T$–$S$ plane, in which the degree of final cooperation exceeds an arbitrarily chosen threshold value of 0.3. In the blue area, this is always the case. In the green region, this holds if the system began with cooperative hubs or a metric cluster. In the yellow region, the cooperative threshold is only exceeded if the system began with cooperative hubs (see Supplementary Fig. 2 for details). This behavior is significantly different in the case of the arXiv collaboration network (Fig. 4b). In contrast to the previous case, there is no region where only initially cooperative hubs allow for sustained cooperation. In the gray region, however, final cooperation only exceeds the threshold value if the system was started with cooperators forming a metric cluster. Hence, whereas in the Internet IPv6 topology hubs can drive the system toward cooperation, in the case of the arXiv network metric clusters are more efficient at sustaining cooperation than the most connected nodes. We observe a similar behavior using synthetic scale-free networks with different mean local clustering coefficients and power-law exponents. Figure 4c shows a similar behavior to that of the Internet (i.e., the hubs are more efficient than metric clusters), whereby the networks were generated with a power-law exponent $\gamma = 2.4$ and mean local clustering coefficient $\bar{c} = 0.5$ (here, cooperation is sustained in none of the cases in the red region). In Fig. 4d, we find a behavior similar to the arXiv (i.e., metric clusters are more efficient than the hubs), where we have generated synthetic networks with power-law exponent $\gamma = 2.9$ and clustering $\bar{c} = 0.6$. To conclude,

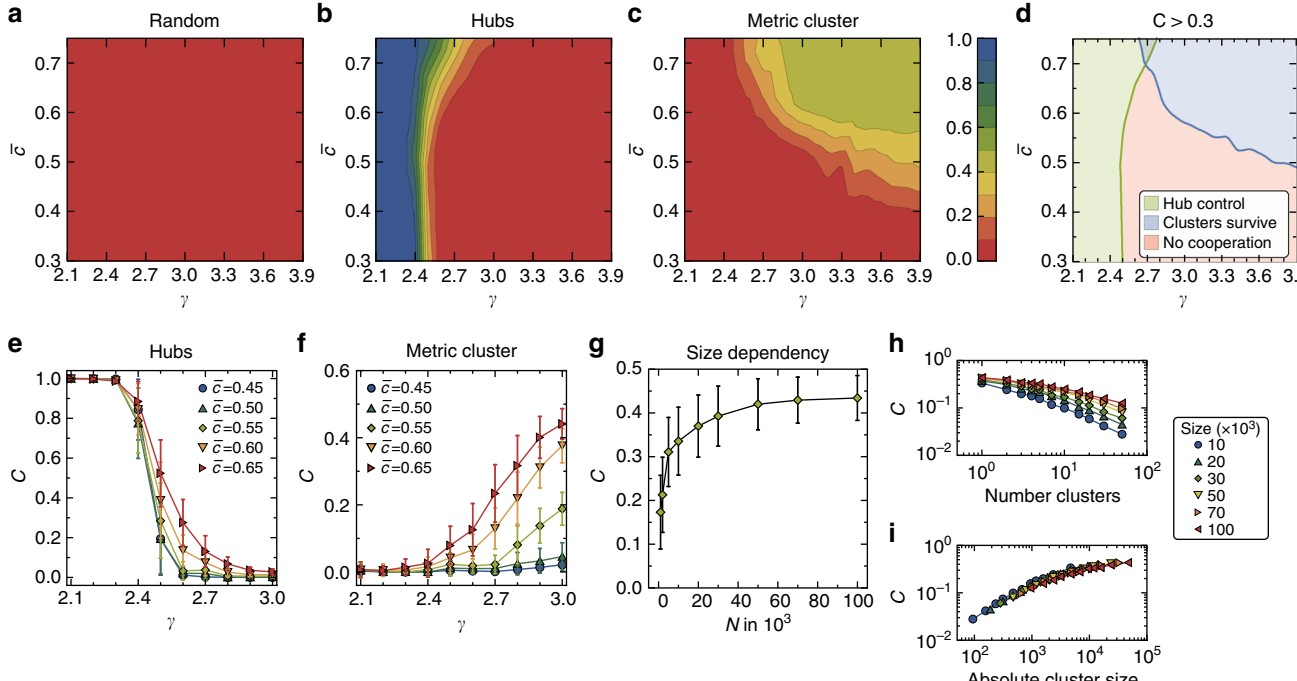

**Fig. 5** The impact of heterogeneity and clustering. **a–c** Final (after $2 \times 10^5$ update steps) density of cooperators (color coded) for the prisoner's dilemma game ($T = 1.5$ and $S = -0.5$) averaged over 50 realizations as a function of the degree distribution power-law exponent $\gamma$ and mean local clustering $\bar{c}$. Networks have $N = 2 \times 10^4$ nodes and a mean degree $\langle k \rangle \approx 6$. The initial density of cooperators is always $c(0) = 0.5$. **a** Randomly assigned initial cooperators. **b** Hubs are assigned as initial cooperators with a probability $p \propto k$. **c** Initial cooperators are localized in the angular space. **d** Regions where the final cooperation exceeds a threshold value of 0.3 for the cases presented in **a–c**. **e** Final cooperation as a function of the network heterogeneity for different values of $\bar{c}$ starting with preferential hub assignment, representing different cuts through **b**. Error bars denote one standard deviation from top to bottom. **f** Final cooperation as a function of the network heterogeneity for different values of $\bar{c}$ starting with cooperators assigned into a metric cluster, representing different cuts through **c**. Error bars denote one standard deviation from top to bottom. **g** Final cooperation density as a function of the network size for synthetic networks with power-law exponent $\gamma = 2.9$, $\bar{c} = 0.6$, and mean degree $\langle k \rangle \approx 6$. Error bars denote one standard deviation from top to bottom. **h** Final cooperation for different network sizes (see legend) and the same parameters as before. Initial cooperators ($c(0) = 0.5$) are assigned into different numbers of disjoint metric clusters, whose number is plotted on the x-axis. **i** The same as **h**, but on the x-axis the resulting absolute size of each cluster is shown, given by the number of nodes divided by twice the number of cooperating clusters

in very heterogeneous networks, hubs are efficient at driving the system towards cooperation, whereas in less heterogeneous—but including scale-free—networks, metric clusters are more efficient.

To investigate this effect in detail, let us now consider the prisoner's dilemma game, in particular parameters $S = -0.5$ and $T = 1.5$ in the payoff matrix from Eq. (1), which is widely used as a proxy for real social dilemma situations. We vary the network topology using the model mentioned earlier. In particular, we tune the heterogeneity in terms of the power-law exponent $\gamma$ and the mean local clustering coefficient, $\bar{c}$, which is a measure of the strength of the underlying metric structure[21]. We consider the different strategies of allocating the initial cooperators discussed before.

The combination of the initial conditions and the network topology yields particularly interesting insights. If the initial cooperators are distributed randomly, final cooperation is always very low for the chosen parameters $T$ and $S$ (Fig. 5a). We find the same result (Supplementary Fig. 5) if the initial cooperators are assigned into a connected (i.e., unique network[18]) cluster (see Methods). However, if the initial cooperators are distributed among the hubs of the system and the network is sufficiently heterogeneous, they are able to drive the system to a highly cooperative state (see blue region Fig. 5b, and Supplementary Movie 2). Large mean local clustering $\bar{c}$, which implies a strong metric structure, adds to this effect (cf. green region in Fig. 5d, e), in agreement with ref. [32]. Importantly, if the network is not

sufficiently heterogeneous, but still scale-free, the hubs lose their ability to control the system and defection eventually prevails (red region in Fig. 5b, Supplementary Movie 3). In contrast, if we begin with the initial cooperators clustered in the metric space, this will allow for sustained cooperation even in scale-free networks, but only if the metric structure is sufficiently strong (see Fig. 5c, blue region in Fig. 5d, and Supplementary Movie 4). If the network becomes too heterogeneous, the clusters are no longer sustained (see Fig. 5f and Supplementary Movie 5).

We also investigate whether network and cluster size affect the ability of metric clusters of cooperators to survive. In Fig. 5g, we show that the final cooperation density increases with the system size and saturates to a value close to $C = 0.5$. For a fixed network size, cooperation decreases if we assign cooperators into a larger number of smaller clusters (see Methods), as shown in Fig. 5h. However, if plotted as a function of the absolute size of the individual clusters (which can be calculated by dividing the number of nodes by twice the number of clusters), the curves that correspond to different network sizes collapse, see Fig. 5i. This suggests that the survival of a metric cluster of cooperators is directly related to its absolute size.

Finally, we can formulate approximate conditions for the survival of cooperating metric clusters. Their survival is favored if they are large enough, i.e., their size is $n_c > 10^3$ (Fig. 5g), if the mean local clustering is high enough, i.e., $\bar{c} > 0.5$ (Fig. 5f), and if the network is not too heterogeneous, i.e., $\gamma > 2.5$.

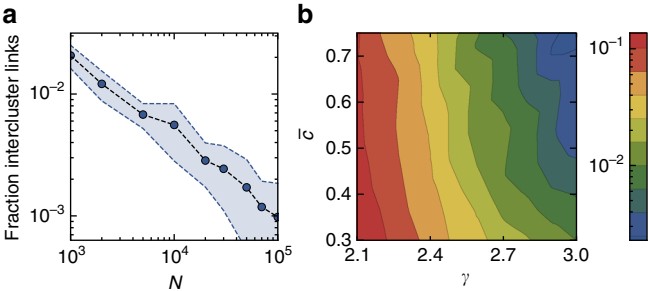

**Fig. 6** Intercluster links. **a** Fraction of links between nodes in a metric cluster spanning half of the network to nodes outside of the cluster compared to the total number of links in the network as a function of the network size for synthetic networks with power-law exponent $\gamma = 2.9$, $\overline{c} = 0.6$, and mean degree $\langle k \rangle \approx 6$. The shaded area denotes one standard deviation from top to bottom. **b** The same fraction of links (color coded) for synthetic networks with $N = 2 \times 10^4$ nodes and $\langle k \rangle \approx 6$ as a function of the power-law exponent $\gamma$ and mean local clustering $\overline{c}$

**Intercluster links explains the survival of metric clusters**. The survival of metric clusters of cooperators can be understood as analogous to spatial selection in Euclidean space in lattice topologies. In this case, clusters of cooperators survive because they are shielded from surrounding defectors, such that the interactions between cooperators and defectors only occur at the border of the clusters. Similarly, in heterogeneous networks, metric clusters survive because they are shielded from surrounding defectors and their spatial organization reduces the number of interactions between cooperating and defecting individuals. For larger clusters, the relative surface area of the border in contact with adjacent defectors decreases, which shields them more effectively and hence explains why they are more likely to survive (Fig. 6a). For a given size, two different mechanisms determine the number of links between spatially clustered cooperators and defectors. Firstly, the greater the degree of heterogeneity, the larger the number of hubs, i.e., high degree nodes. These nodes are connected to many other nodes, and therefore form long-range connections in the metric representation, which are likely to connect cooperators and defectors. This is the reason why more heterogeneity hinders the survival of metric clusters (Fig. 6b). For a fixed level of heterogeneity, increasing the mean local clustering coefficient will reduce the temperature $\overline{T}$, which reduces the amount of long-range connections due to randomness, cf. Fig. 1a, b. Therefore, a higher degree of mean local clustering reduces the number of intercluster links, which in turn favors the survival of metric clusters as explained before (Fig. 6b).

## Discussion

Structured populations play an important role in the evolution of cooperation in social dilemmas. Real contact networks are heterogeneous (often scale-free) and exhibit a high mean local clustering coefficient. The latter implies the existence of an underlying geometry[21]. Specifically, real heterogeneous networks can be embedded into hyperbolic space comprising a popularity (radial) dimension and a similarity (angular) dimension.

We have shown that this underlying metric space plays an important role in the evolution of cooperation in heterogeneous contact networks. Specifically, evolutionary dynamics lead to the formation of clusters of cooperators in the angular dimension of the underlying metric space, akin to spatial selection in Euclidean space[13, 14]. This behavior can be understood in terms of the fraction of intercluster links that determines how well metric clusters of cooperators are shielded from surrounding defectors. Depending on the power-law exponent $\gamma$ of the degree

distribution and the mean local clustering coefficient $\overline{c}$ (which is proportional to the strength of the metric structure), metric clusters can be more efficient at sustaining cooperation than the most connected nodes, which is the case in the arXiv collaboration network. Only when the network is very heterogeneous, such as in the case of the Internet IPv6 topology, are hubs more effective at promoting cooperation. We have shown that if cooperators are clustered in the metric space, heterogeneity can hinder cooperation in the prisoner's dilemma. Finally, one could argue that such a configuration is more realistic than random initial conditions, as for example the nodes in the Internet network that correspond to the same countries are naturally clustered in the metric space (see ref. [25], and different countries adopt different attitudes towards mitigating climate change[33].

Our findings reveal that heterogeneity does not always favor cooperation in evolutionary games on structured populations, but can even have the opposite effect, thus complementing existing studies about the impact of heterogeneity of realistic contact networks. Furthermore, our framework unifies the description of spatial effects and the heterogeneity of contact networks. This framework can be applied to different games and extended to multiplex networks, opening promising future lines of research.

## Methods

**Evolutionary game dynamics**. In the evolutionary game dynamics considered here, individuals play strategic games with their contacts where, for instance, they have two strategic choices: they can either cooperate (C) or defect (D). The payoff of each two-player game is then described by the payoff matrix

$$M = \begin{array}{c|cc} & C & D \\ \hline C & 1 & S \\ D & T & 0 \end{array}. \tag{1}$$

Parameters $T$ and $S$ define different games[6]. $T < 1$ and $S > 0$ defines the "harmony" game, $T < 1$ and $S < 0$ corresponds to the "stag hunt" game, $T > 1$ and $S < 0$ yields the "prisoner's dilemma", and finally for $T > 1$ and $S > 0$ we obtain the "snowdrift" game.

One round of the game consists of each individual playing one game with each of her neighbors in the network of contacts. For each game, nodes collect payoffs given by Eq. (1), which depend on the strategies of the involved players. Here we consider the evolution of the system to be governed by imitation dynamics[34–36] (Fig. 1c), reflecting that individuals tend to adopt the strategy of more successful neighbors. After each round of the game (synchronous updates) each node $i$ chooses one neighbor $j$ at random and copies her strategy with probability $P_{i \leftarrow j}$, specified by the Fermi–Dirac distribution[15, 37, 38]

$$P_{i \leftarrow j} = \frac{1}{1 + e^{-(\pi_j - \pi_i)/K}}, \tag{2}$$

motivated by maximum entropy principles in Glauber-like dynamics[34, 39]. $\pi_i$ and $\pi_j$ measure the payoffs of nodes $i$ and $j$, while $K$ denotes the irrationality of the players, which we set to 0.5. After all nodes have updated their strategy simultaneously, we reset all payoffs.

In this contribution, games are played only on the giant connected component (GCC) of the network of contacts.

**Complex networks embedded into underlying metric spaces**. Metric spaces underlying complex networks provide a fundamental explanation of their observed topologies[23, 24]. In the class of models used here, each node $i$ is mapped into the hyperbolic disc where it is represented by the polar coordinates $r_i, \theta_i$. These coordinates abstract the popularity and similarity of nodes[24]. The radial coordinate $r_i$ is related to the expected degree of node $i$ and therefore abstracts its popularity. More popular nodes are located closer to the center of the disc (lower radial coordinate). The angular distance between nodes $i$ and $j$, $\Delta \theta_{ij} = \pi - |\pi - |\theta_i - \theta_j||$, is an abstract measure of their similarity. Lower distance implies higher similarity. The hyperbolic distance[23]

$$x_{ij} = \cosh^{-1}\left(\cosh r_i \cosh r_j - \sinh r_i \sinh r_j \cos \Delta \theta_{ij}\right), \tag{3}$$

combines information about both popularity and similarity of nodes $i$ and $j$, such that the connection probability for a given pair of nodes depends only on their hyperbolic distance.

To generate networks based on hidden hyperbolic space, we distribute nodes on the hyperbolic disc by assigning polar coordinates $(r_i, \theta_i)$ to each node. In particular, we draw $\theta_i$ from the uniform distribution $\mathcal{U}_{[0,2\pi)}$ and radial coordinates

$r_i$ from the distribution

$$\rho(r) = \frac{1}{2}(\gamma - 1)e^{\frac{1}{2}(\gamma-1)(r-R)}, \tag{4}$$

where $R$ denotes the disc radius given by[23]

$$R = 2\ln\left[\frac{2TN}{\bar{k}\sin\bar{T}\pi}\left(\frac{\gamma-1}{\gamma-2}\right)^2\right], \tag{5}$$

where $N$ denotes the number of nodes, $\gamma$ is the power-law exponent of the degree distribution, and $T$ denotes the temperature. Finally, we connect pairs of nodes $i$ and $j$ with probability $p(x_{ij})$, which depends exclusively on the hyperbolic distance $x_{ij}$ between nodes $i$ and $j$. The connection probability is given by the Fermi–Dirac distribution

$$p(x_{ij}) = \frac{1}{1 + e^{\frac{1}{2T}(x_{ij}-R)}}, \tag{6}$$

where the aforementioned temperature $\bar{T}$ controls the strength of the metric structure and the level of mean local clustering, $\bar{c}$. This is illustrated in Fig. 1a, b.

Finally, given a real network, coordinates of the nodes can be inferred using maximum likelihood estimation techniques[40, 41]. This enables us to identify the set of coordinates that maximize the probability that the observed real-world network was generated using the described model. The inferred hyperbolic maps have proven to be very accurate in the case of scale-free, clustered networks[25, 42, 43].

**Mean local clustering and relation to the spatial structure**. The local clustering coefficient of node $i$ is defined as[44]

$$c_i = \frac{2 \cdot \text{Number of closed triangles } i \text{ participates in}}{k_i(k_i - 1)}, \tag{7}$$

where $k_i$ denotes the degree of node $i$. The maximal number of closed triangles a node with degree $k_i$ can participate in is $k_i(k_i - 1)/2$. The mean local clustering coefficient of a given network is then the average of $c_i$ over all nodes with $k > 1$ (nodes with $k = 1$ cannot participate in any triangles).

In the framework introduced in the previous section, a low temperature $\bar{T}$ implies a high mean local clustering, which is the consequence of the triangle inequality in the underlying metric space (Fig. 1a). A high temperature, however, induces more randomness in the form of long-range connections (see Eq. (6)), which reduces the mean local clustering coefficient (Fig. 1b). See ref. [23] for further details.

**KS statistic**. The KS statistic[28], which we denote as $\bar{\rho}$, is defined as the maximum absolute difference between the values of two cumulative distributions. We are interested in measuring the difference between the distribution of cooperators in the angular space, $c(\theta)$, whose cumulative distribution is given by $C(\theta) = \int_0^\theta d\theta' c(\theta')$, and the uniform distribution. Then, the KS statistic is given by

$$\rho_C = \max_{\theta \in \{0, 2\pi\}} \left| C(\theta) - \frac{\theta}{2\pi} \right| \tag{8}$$

and analogously

$$\rho_D = \max_{\theta \in \{0, 2\pi\}} \left| D(\theta) - \frac{\theta}{2\pi} \right|, \tag{9}$$

where $D(\theta) = \theta - C(\theta)$ denotes the cumulative distribution of defectors in the angular space. Finally, we define

$$\bar{\rho} = c\rho_C + (1 - c)\rho_D, \tag{10}$$

where $c$ denotes the density of cooperators at the current timestep. Note that here we omitted the time dependency.

**Assignment of initial cooperators**. In this contribution, we always start with an initial cooperation density of $C(0) = 0.5$. However, the distribution of initial cooperators in the network can be different. In particular, we distinguish between the following procedures.

Random assignment: Each node is initialized as a cooperator with 50% probability and as a defector otherwise.

Hubs: We preferentially assign hubs as initial cooperators. To this end, we assign $N/2$ cooperators which we select proportional to their degree, i.e., $p_c(k) \propto k$. $N$ denotes total number of nodes in the network.

Metric cluster: We sort all nodes by their angular coordinate $\theta$, and assign the first $N/2$ nodes as cooperators.

Multiple metric clusters: We again sort all nodes by their angular coordinate $\theta$. We now fix a number of distinct clusters, $n_c$ and assign the first $N/(2n_c)$ nodes as cooperators, the second $N/(2n_c)$ nodes as defectors, the third $N/(2n_c)$ as cooperators and so on. See Supplementary Fig. 6 for an explicit example.

Connected cluster: We assign $N/2$ nodes into a connected cluster, or unique network cluster[18]. To this end, we start from the initial graph and randomly remove nodes until the size of the giant connected component (GCC) reaches $N/2$. The nodes that are now in the GCC are assigned as cooperators in the original graph, and the remaining $N/2$ nodes are assigned as defectors. This procedure ensures that the initial cooperators form a unique connected cluster. Note that a network cluster in general is not the same as a metric cluster. Instead of $N/2$, we can use any size of the connected cluster, for example in Fig. 3 we used $N/4$.

**Empirical networks**. The arXiv data are taken from ref. [45] and contains co-authorship networks from the free scientific repository arXiv. The nodes are authors that are connected if they have co-authored a paper. In arXiv, each paper is assigned to one or more relevant categories. The data only cover papers containing the word "networks" in the title or abstract from different categories up to May 2014. Here we consider the category "Molecular Networks" (q-bio.MN). The network has ~1905 nodes, mean degree $\langle k \rangle = 4.6$, clustering coefficient $\bar{c} = 0.66$, and a power-law degree exponent $\gamma = 3.9$.

The IPv6 Autonomous Systems (AS) Internet topology was extracted from the data collected by the Archipelago active measurement infrastructure (ARK) developed by CAIDA[46]. The connections in each topology are not physical but logical, representing AS relationships. An AS is a part of the Internet infrastructure administrated by a single company or organization. Pairs of ASs peer to exchange traffic. These peering relationships in the AS topology are represented as links between AS nodes. CAIDA's IPv6 data sets provide regular snapshots of AS links derived from ongoing traceroute-based IP-level topology measurements (data sets are available at http://www.caida.org/data/active/ipv4_routed_topology_aslinks_dataset.xml and https://www.caida.org/data/active/ipv6_allpref_topology_dataset.xml). The considered topology was constructed by merging the AS link snapshots during the first 15 days of January 2015, which are provided at http://data.caida.org/datasets/topology/ark/ipv6/as-links/2015/01/ and http://data.caida.org/datasets/topology/ark/ipv4/as-links/. The network consists of $N = 5162$ nodes, has a power law degree distribution with exponent $\gamma = 2.1$, average node degree $\bar{k} = 5.2$, and average clustering $\bar{c}_2 = 0.22$. The hyperbolic maps for both data sets have been taken from ref. [43].

Supplementary Figure 2 shows the final density of cooperators for both networks and for the different allocation strategies described in the main text.

**Data availability**. The empirical data sets as well as an implementation of the model networks used in this study have been made available at https://figshare.com/articles/DataAndModel_zip/4817947. An implementation of the technique to construct hyperbolic maps for real networks[40, 41] is publicly available at https://bitbucket.org/dk-lab/2015_code_hypermap. Any additional data that support the findings of this study are available from the corresponding author upon reasonable request.

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

## Acknowledgements
We thank Stefano Duca for interesting and helpful discussions. We thank Heinrich Nax for feedback on the manuscript. We thank Eoin Jones for proofreading the manuscript. K.-K. K. acknowledges support by the ERC Grant "Momentum" (324247).

## Author contributions
K.-K.K. designed the research, conducted the research, performed all calculations and numerical simulations, created the figures and wrote the manuscript.

## Additional information

**Competing interests:** The authors declare no competing financial interests.

