## [Peer Review File · Nature Communications]

Reviewers' comments:

Reviewer #1 (Remarks to the Author):

The study suggests the presence of spatial selection (one of the mechanisms behind the evolution of cooperation in structured populations) in evolutionary games on scale-free networks. Considering both real and synthetic networks with an hidden metric structure, the author finds that cooperators form clusters in this metric space and that they are evolutionary stable. Also, using different initial conditions, he suggests that spatial selection is more effective than network heterogeneity in sustaining cooperation. The major claims of the work are:

- Spatial selection in scale-free networks leads to the formation of clusters of cooperators in the metric space that can resist the invasion of defectors. Those clusters are evolutionary stable.

- Spatial selection can be more efficient in sustaining cooperation than network heterogeneity.

- In some cases, network heterogeneity can hinder cooperation.

Technically speaking, the study seems correct and the claims, except for one point (see below), are supported by the analysis proposed by the author. However, there are important logical flaws in the analysis and an important phenomenon well known in the literature has not been considered. Thus, my major concerns are about the novelty and the importance of the claims along with some methodological issues.

Major concerns:

1. Spatial selection, as introduced by Nowak and May (ref. 17 in the manuscript), refers to the formation of clusters of cooperators able to survive the invasion of defectors and thus sustain cooperation. The formation of those clusters is correlated with local clustering coefficient as cohesive groups of nodes have more chances to form such clusters than less cohesive ones. The

main claim of the study is the formation of those clusters also in the metric space and thus the presence of spatial selection even in scale-free networks. The main reasoning behind it, as stated in the introduction, is: “a high clustering coefficient implies the existence of an underlying metric space (16), which in turn allows for spatial selection”.

The formation of cooperative clusters, and their evolutionary stability, is well known in the literature under the name of network reciprocity. Actually, in his seminal paper of 2006 “Five Rules for the Evolution of Cooperation”[1] Nowak suggests that network reciprocity is a generalization of spatial reciprocity:

“The individuals of a population occupy the vertices of a graph. The edges determine who interacts with whom. Let us consider plain cooperators and defectors without any strategic complexity. A cooperator pays a cost, c , for each neighbor to receive a benefit, b . Defectors have no costs, and their neighbors receive no benefits. In this setting, cooperators can prevail by forming network clusters, where they help each other. The resulting network reciprocity is a generalization of spatial reciprocity (40).”

Where (40) is the same paper of ref. 17 of the manuscript. In this sense, network reciprocity is the extension of spatial selection from a metric space to the non-metric space of the network. In spatial selection cooperators close in the metric space form clusters that are able to resist the invasion of defectors. In the same way, in network reciprocity cooperators close in the network (neighbors or communities) form similar clusters.

Along the same line there is also [2] (ref. 13 in the manuscript), where the authors study the formation of cooperative clusters in poissonian and scale-free networks finding that, in scale free networks, cooperators survive forming a unique growing cluster. That is exactly what the author sees in Figs.2A-B.

In the present work, the author maps network reciprocity back in a metric space finding that clusters of cooperators that are close in the network are also close in the hyperbolic space. But this is a trivial result of the mapping. As explained in the methods section, nodes close in the hyperbolic space have a higher probability of being connected. In this way, nodes connected in the network are probably mapped in the same region of the space. Thus a cluster of cooperators in the network is very likely the same cluster in the metric space.

2. Another important flaw of the work is that network reciprocity is not even mentioned. This suggests that heterogeneity is the main mechanism behind cooperation in scale-free networks (i.e.

see the second sentence of the abstract) when actually both play an important role (network reciprocity is probably even more important, e.g. we see cooperation also in homogenous graphs). This profoundly alters the interpretation of the results and the conclusions.

I.e. the second major claim of the work is that spatial selection is more effective than heterogeneity in sustaining cooperation. Also in this case, if we frame spacial selection as network reciprocity this is a somewhat expected result. Local clustering plays an important role in sustaining cooperation (i.e. [3]) as it favours the formation of clusters of cooperations. In the analysis of Fig. 3 the author considers a highly heterogeneous network with low clustering (IPv6 network $\gamma = 2.6$ and $c = 0.22$) and a less heterogeneous one with a sensibly higher clustering (arXiv collaborations $\gamma = 2.9$ and $c = 0.66$). Given the lower heterogeneity and the higher clustering of the second network it is highly probable that network reciprocity will play a more important role in that network with respect to the first one even without running the simulations.

3. Following on the same line, there are a couple of methodological points that probably should be clarified.

- When studying the effect of initial conditions the role of degree correlations are not taken into account and they could sensibly alter the results, i.e. [4]. Actually the two real networks considered are the archetypal examples of disassortative (IPv6 AS) and assortative networks (arXiv collaboration). See table 1 of [5].

- The same applies for the synthetic networks used in the analysis of Figs. 4 and 5. For high heterogeneity some correlations could be present that could disappear for larger values of γ .

Thus, to support the third claim more tests taking into account all the topological features that can alter the results are needed.

Minor points:

4. In the methods, in the “Evolutionary game dynamics” section: the word “repetitive” can be misleading for the reader as it suggests repeated games while the paper deals with single-shot games.

5. Same section. At the end of the first paragraph and, at the beginning of the second: Fig.1A is cited. Probably it should be Fig. 1B.

6. First column, third line of page 7: in this case should be Fig.1A instead of Fig.1B. The same applies for line 11 of the first column of page 2.

7. Sixth line of the first column of page 2: abstracts instead of abstract.

8. Caption of Fig.1 of the SI. there is a missing reference “??”.

Concluding, the intuition that the metric space should lead to spatial selection is interesting but the study should be entirely rethought and reframed to take into account network reciprocity. This will surely affect the importance and novelty of the results.

Refs.

1. Science 314(5805):1560-1563, 2006.
2. Physical Review Letters 98, 108103, 2007.
3. Physical Review E 78, 017101, 2008.
4. Physical Review E 76, 027101, 2007.
5. Physical Review Letters 89, 208701, 2002.

Reviewer #2 (Remarks to the Author):

Major comments:

What are the major claims of the paper?

The major claim of the paper is that network heterogeneity (as described by the degree distribution's power law slope) does not always favour the emergence of cooperation in social dilemmas played on the network, but can instead be overridden by the effect of the local clustering structure of the network, which the authors encode as an underlying metric space via recent hyperbolic embedding methods.

Are they novel and will they be of interest to others in the community and the wider field? If the conclusions are not original, it would be helpful if you could provide relevant references.

Although the general points made by the paper (about the role of heterogeneity and clustering) are not necessarily very original (see for example Assenza, Salvatore, Jesús Gómez-Gardeñes, and Vito Latora. "Enhancement of cooperation in highly clustered scale-free networks." *Physical Review E* 78.1 (2008): 017101; Perc, Matjaž, et al. "Evolutionary dynamics of group interactions on structured populations: a review." *Journal of the royal society interface* 10.80 (2013): 20120997.), the paper shines a new light on these observations from the perspective of a competition between local structure and global heterogeneity. This is a novel and very interesting contribution, in addition to the observation that heterogeneity can be detrimental to cooperation in certain settings.

Is the work convincing, and if not, what further evidence would be required to strengthen the conclusions? On a more subjective note, do you feel that the paper will influence thinking in the field? Please feel free to raise any further questions and concerns about the paper.

I think the paper makes an interesting point —that we find a trace of spatial selection in a new space derived from the network— and that in cases of strongly clustered structures, the heterogeneity of the network can hinder cooperation. What I think is still missing in the picture drawn from this work is an understanding (and possibly) quantification of what really is happening in the network and what are the main ingredients that control whether heterogeneity is good/bad for cooperation, what level of local cluster is required for cooperation to be stable etc.

For example, the author notes that when the metric structure is strong the heterogeneity can be harmful to cooperation. Why does this happen? is it because the hubs existing behave as bridges between (hyperbolic)spatially separated regions hence promoting mixing and destabilising cooperator clusters? Is it because there are different dynamics happening at the interfaces between clusters? Or does that happen due to a different mechanism?

Secondly, although I personally like very much the work related to the extraction of underlying spaces for networks, I wonder how much this abstraction can and will be received by the general

community working in evolutionary game theory. This is due to a mixture of technical expertise required to understand the methods and of a certain lack of direct interpretation implicit in these embeddings. I imagine this will be a very popular paper with the stat-mech/network community, but I have a hard time gauging the potential impact in others.

We would also be grateful if you could comment on the appropriateness and validity of any statistical analysis, as well the ability of a researcher to reproduce the work, given the level of detail provided.

I have one question/doubt regarding the measure of cluster timescale, which is not likely to change the final picture but I think it is important to raise. The author adopts a KS test between the angular distribution of cooperators/defectors and then weighs it with the cooperators fraction to extract a measure used to define the stability of the state.

He mentions that the cooperator distribution is tested against a normal distribution, although I think he meant the uniform one (as per Eq. 8). Now my point is that I'm confused: if we use a normal distribution for the angular distribution of cooperators, I imagine that prior to mean that we are looking for a coop cluster that is localised in angular space; on the contrary, if we use the uniform distribution (which I deem more correct in this case) the measure is averaging over all the whole angular structure of the network, and measuring any large localisation of cooperators. However, this assumption implies that we consider the clustered state as the state of the whole network, rather than track the individual evolution of various spatial clusters (which instead really are the focus of the paper). Also, this type of measure has problems distinguishing configurations that have moving but fixed-density distributions of cooperators (e.g. a cluster that evolves rotating around the network would always have the same ρ , but it would be a very different cluster at each time).

How much would it change if one adopted something along the lines of KL divergence between configurations at different times?

Regarding reproducibility, the author provided the data and pointers to the code needed to reproduce the paper (potentially on other network datasets), so the work is definitely reproducible by other researchers (although due to the nature of the embedding construction, it might not be very easy for researchers outside of physics/math/compsci fields).

Overall assessment:

Based on the comments above, I consider this paper as a solid and interesting contribution. However, I harbour doubts about its potential impact to the broader community in consideration of the elements of interpretation that I highlighted above. So I would recommend a revision of the manuscript to address these concerns before reaching a final decision.

*Minor comments: *

- The paper is a bit sloppily edited in some respects; I found a few typos and grammatical stumbles now and there ("haven", second column page 1, "ordinate abstract" first column page 2, "spacial" second column page 4, the panels B and C in the caption of Figure 4 seem to have been swapped, there are broken Latex \backslash ref in the captions of a few figures in the SI)

Reviewer #3 (Remarks to the Author):

Comments on "Spatial selection on scale-free networks"

This paper studied the evolution of cooperation on social networks, which is one of the most popular topics in evolution game. Many theoretical studies have found that network heterogeneity favors cooperation. One of the main reasons behind the increase of cooperation in heterogeneous networks is that hubs are usually occupied by cooperators, which ensures their long-term success in the evolutionary process. In this paper, however, the author indicated that "spatial selection" can be even more efficient than heterogeneity in sustaining cooperation. Frankly speaking, I don't understand why network heterogeneity is not a part of "spatial selection". After reading the paper, I guess the author wants to say that if the local clustering coefficient in scale-free networks is large, then cooperators can survive better in local clusters than hubs.

As a general assessment, this paper has the potential of making a nice contribution to this field (i.e., how the local clustering coefficient affects cooperation). However, it has several problems. Specifically, I have two main comments, why clustering coefficient is important, and whether this result holds in general.

(1) In the discussion session, the author only summarizes the simulation outcomes, but did not discuss why and how the local clustering coefficient works. I think the author should provide a reasonable explanation for the main result. A possible choice would be to give some toy examples (see examples in Allen et al., 2017) to show that sometimes the hub cannot maintain cooperation but clusters can. This will help us to understand the role of local structure in promoting cooperation. Furthermore, a carefully discussion for the main result is necessary. In fact, it is counter intuitive that the hubs defect but cooperation can be stably maintained in leaves, because in this case the hubs will have higher payoffs than the leaves and the leaves will then imitate the hubs. Sanots et al.,

(2008) have provided a very good example (double-star graph) to demonstrate why the hub is crucial in promoting cooperation. So I think you should also give some examples to support your result.

(2) The second problem is whether this result holds in general. The author observed that under certain conditions, spatial selection becomes more important than heterogeneity. However, these observations are based on numerical simulations with some specific parameters or networks. Thus, the universality of this result is questionable. I do realize that it is very difficult to provide a rigorous mathematical analysis. But at least, the author should analyze some toy examples carefully, and derive the condition for these examples.

Other issues:

1. The definition of “spatial selection” is confusing. From my point of view, “spatial selection” introduced by Nowak and May (1992) means that there is a spatial structure in the evolutionary process. The spatial structure can be homogenous or heterogeneous, and spatial clusters can be formed in both homogenous and heterogeneous networks. In particular, the hub can also form spatial clusters (e.g., double-star graph in Santos et al., 2008). Thus, I don’t understand why the author thinks spatial selection and network heterogeneity are two different things.

2. Section B is based on the stag hunt game with $T=0.5$ and $S=-0.5$. However, this game is not really a social dilemma. With these parameters, cooperation and defection is bistable, the mixed equilibrium is at $1/2$, and cooperation is payoff dominance. That is, cooperation should be better than defection. Thus, it is strange for me that the cooperation level is only 0.4 after 10^5 update steps. If you run the simulation for a longer time, I guess the cooperation level will finally converge to 1.

3. Continue to the above issue. It is better to start with the prisoner’s dilemma game in Section B. For me, Figure 4 is more interesting than Figure 2.

References:

Allen, B., Lippner, G., Chen, Y. T., Fotouhi, B., Momeni, N., Yau, S. T., & Nowak, M. A. (2017). Evolutionary dynamics on any population structure. *Nature*, 544(7649), 227-230.

Santos, F. C., Santos, M. D., & Pacheco, J. M. (2008). Social diversity promotes the emergence of cooperation in public goods games. *Nature*, 454(7201), 213-216.

Revised manuscript “Metric clusters in evolutionary games on scale-free networks”

Response to Referees’ Comments

Referee B: see page 4

Referee C: see page 7

Referee A:

The study suggests the presence of spatial selection (one of the mechanisms behind the evolution of cooperation in structured populations) in evolutionary games on scale-free networks. Considering both real and synthetic networks with an hidden metric structure, the author finds that cooperators form clusters in this metric space and that they are evolutionary stable. Also, using different initial conditions, he suggests that spatial selection is more effective than network heterogeneity in sustaining cooperation. The major claims of the work are:

- Spatial selection in scale-free networks leads to the formation of clusters of cooperators in the metric space that can resist the invasion of defectors. Those clusters are evolutionary stable.*
- Spatial selection can be more efficient in sustaining cooperation than network heterogeneity.*
- In some cases, network heterogeneity can hinder cooperation.*

Technically speaking, the study seems correct and the claims, except for one point (see below), are supported by the analysis proposed by the author. However, there are important logical flaws in the analysis and an important phenomenon well known in the literature has not been considered. Thus, my major concerns are about the novelty and the importance of the claims along with some methodological issues.

Authors’ response: We thank the referee for his/her time reviewing our manuscript and for the valuable feedback, that has helped to improve the manuscript substantially. We hope that the improvements will convince the referee of the novelty of our work. We provide a detailed answer to each point raised in the following.

1. Spatial selection, as introduced by Nowak and May (ref. 17 in the manuscript), refers to the formation of clusters of cooperators able to survive the invasion of defectors and thus sustain cooperation. The formation of those clusters is correlated with local clustering coefficient as cohesive groups of nodes have more chances to form such clusters than less cohesive ones. The main claim of the study is the formation of those clusters also in the metric space and thus the presence of spatial selection even in scale-free networks. The main reasoning behind it, as stated in the introduction, is: “a high clustering coefficient implies the existence of an underlying metric space (16), which in turn allows for spatial selection”. The formation of cooperative clusters, and their evolutionary stability, is well known in the literature under the name of network reciprocity. Actually, in his seminal paper of 2006 “Five Rules for the Evolution of Cooperation”[1] Nowak suggests that network reciprocity is a generalization of spatial reciprocity: “The individuals of a population occupy the vertices of a graph. The edges determine who interacts with whom. Let us consider plain cooperators and defectors without any strategic complexity. A cooperator pays a cost, c , for each neighbor to receive a benefit, b . Defectors have no costs, and their neighbors receive no benefits. In this setting, cooperators can prevail by forming network clusters, where they help each other. The resulting network reciprocity is a generalization of spatial reciprocity (40).” Where (40) is the same paper of ref. 17 of the manuscript. In this sense, network reciprocity is the extension of spatial selection from a metric space to the non-metric space of the network. In spatial

selection cooperators close in the metric space form clusters that are able to resist the invasion of defectors. In the same way, in network reciprocity cooperators close in the network (neighbors or communities) form similar clusters.

Along the same line there is also [2] (ref. 13 in the manuscript), where the authors study the formation of cooperative clusters in poissonian and scale-free networks finding that, in scale free networks, cooperators survive forming a unique growing cluster. That is exactly what the author sees in Figs.2A-B.

In the present work, the author maps network reciprocity back in a metric space finding that clusters of cooperators that are close in the network are also close in the hyperbolic space. But this is a trivial result of the mapping. As explained in the methods section, nodes close in the hyperbolic space have a higher probability of being connected. In this way, nodes connected in the network are probably mapped in the same region of the space. Thus a cluster of cooperators in the network is very likely the same cluster in the metric space.

Authors' response: We thank the referee for the important point. Indeed, spatial selection has been discussed for lattice topologies in Euclidean space, and network reciprocity can be considered a generalization of this mechanism to the general case of static networks of contacts. We have added an explanation regarding network reciprocity in the revised version of the manuscript.

The referee refers to the findings in [PRE 78:017101] that cooperators form a unique connected cluster. However, this does not provide further information about the spatial organization of the cooperators (apart from them being connected), nor does the definition of network reciprocity cited by the referee. Importantly, the fact that cooperators form a unique connected cluster does not imply that they form a metric cluster. We acknowledge that this point was not clear in the previous version of the manuscript and we now present additional results where we assign the initial cooperators into a unique connected cluster (the procedure is described in detail in the Methods section of the revised manuscript). These initial conditions lead to a completely different behavior compared to metric clusters (cf. Fig. 4C in the main manuscript and Fig. S4 in the Supplementary Materials). This means that our findings are not a trivial result from the mapping, but indeed offer new insights into the evolution of cooperation in structured populations. The reason for this is that the metric space encodes information about the similarity of nodes that goes beyond their immediate neighborhood.

Furthermore, the mapping allows us to provide a quantitative explanation for the survival of metric clusters similar to spatial selection (see new Sec. IID in the revised manuscript).

2. Another important flaw of the work is that network reciprocity is not even mentioned. This suggests that heterogeneity is the main mechanism behind cooperation in scale-free networks (i.e. see the second sentence of the abstract) when actually both play an important role (network reciprocity if probably even more important, e.g. we see cooperation also in homogenous graphs). This profoundly alters the interpretation of the results and the conclusions.

I.e. the second major claim of the work is that spatial selection is more effective than heterogeneity in sustaining cooperation. Also in this case, if we frame spacial selection as network reciprocity this is a somehow excepted result. Local clustering plays an important role in sustaining cooperation (i.e. [3]) as it favours the formation of clusters of cooperations. In the analysis of Fig. 3 the author consider a highly heterogeneous network with low clustering (IPv6 network $\gamma = 2.6$ and $c = 0.22$) and a less heterogenous one with a sensibly higher clustering (arXiv collaborations $\gamma = 2.9$ and $c=0.66$). Given the lower heterogeneity and the higher clustering of the second network is highly probable that network reciprocity will play a more important role in that network with respect to the first one even without running the simulations.

Authors' response: We again thank the referee for this valuable comment. In the revised version of the manuscript we now discuss network reciprocity and its relation to spatial selection in Euclidean space. We have rephrased several parts of the manuscript to make the distinction clear.

In [PRE 78, 017101], which is Ref [33] in the revised manuscript, the authors find that clustering helps cooperation, which agrees with our findings when we start with cooperative hubs (Fig. 4E in the revised manuscript).

The definition of network reciprocity from [Science 314, 5805:1560], "...cooperators can prevail by forming network clusters... the resulting network reciprocity is a generalization of spatial reciprocity", does not explain how the network clusters are organized nor which network topologies (heterogeneity, clustering...)

particularly favor cooperation. So far, it has been claimed many times that heterogeneity favors cooperation as compared to homogeneous networks (see for instance [PRL 95, 9:098104, PNAS 103, 9:3490]). Here, we show that more heterogeneity can actually hinder cooperation, which is a novel result to our best knowledge, and cannot be explained by network reciprocity alone. However, it can be explained by the fraction of intercluster links that allow metric clusters to survive (see new Sec. IID of the revised manuscript).

We acknowledge that this point was not appropriately covered in the previous version of the manuscript and hope that the new section and the new results in the revised manuscript will be convincing to the referee.

3. Following on the same line, there are a couple of methodological points that probably should be clarified.

- When studying the effect of initial conditions the role of degree correlations are not taken into account and they could sensibly alter the results, i.e. [4]. Actually the two real networks considered are the archetypal examples of disassortative (IPv6 AS) and assortative networks (arXiv collaboration). See table 1 of [5].

- The same applies for the synthetic networks used in the analysis of Figs. 4 and 5. For high heterogeneity some correlations could be present that could disappear for larger values of γ . Thus, to support the third claim more tests taking into account all the topological features that can alter the results are needed.

Authors' response: We again thank the referee for this valuable suggestion. Indeed, the referee is right that the arXiv network is strongly assortative ($r = 0.24$), whereas the Internet network is disassortative ($r = -0.29$). However, the assortativity coefficient of the synthetic networks does not vary significantly in the parameter region of interest, where we have $r \in (-0.05, 0)$ (see the green region in Fig. S1 in the Supplementary Materials).

We have added a section to the Supplementary Materials that explains this important point.

Minor points:

4. In the methods, in the "Evolutionary game dynamics" section: the word "repetitive" can be misleading for the reader as it suggests repeated games while the paper deals with single-shot games.

5. Same section. At the end of the first paragraph and, at the beginning of the second: Fig.1A is cited. Probably it should be Fig. 1B.

6. First column, third line of page 7: in this case should be Fig.1A instead of Fig.1B. The same applies for line 11 of the first column of page 2.

7. Sixth line of the first column of page 2: abstracts instead of abstract.

8. Caption of Fig.1 of the SI. there is a missing reference "??".

Authors' response: We especially thank the referee for pointing out these minor points and have corrected them in the revised version of the manuscript.

Concluding, the intuition that the metric space should lead to spatial selection is interesting but the study should be entirely rethought and reframed to take into account network reciprocity. This will surely affect the importance and novelty of the results.

Authors' response: We have reframed the manuscript following the valuable suggestions of the referee. Most importantly, we now show that metric clusters and connected clusters behave fundamentally different and provide a quantitative explanation for the survival of metric clusters (new Sec. IID in the revised manuscript). We have reframed the term "spatial selection" and its relation to network reciprocity, have conducted a significantly more comprehensive analysis including different network and cluster sizes, and have made an additional effort to make the effect of metric spaces more intuitive for a broad audience (new Figs. 1A and B in the revised manuscript).

Finally, we again thank the referee for his/her helpful comments and for finding our work interesting. We hope to have met his/her expectations with our answers and the revision of the manuscript.

Referee B:

Major comments: *What are the major claims of the paper? The major claim of the paper is that network heterogeneity (as described by the degree distribution's power law slope) does not always favour the emergence of cooperation in social dilemmas played on the network, but can instead be overridden by the effect of the local clustering structure of the network, which the authors encode as an underlying metric space via recent hyperbolic embedding methods.

Are they novel and will they be of interest to others in the community and the wider field? If the conclusions are not original, it would be helpful if you could provide relevant references. Although the general points made by the paper (about the role of heterogeneity and clustering) are not necessarily very original (see for example Assenza, Salvatore, Jesús Gómez-Gardeñes, and Vito Latora. "Enhancement of cooperation in highly clustered scale-free networks." *Physical Review E* 78.1 (2008): 017101; Perc, Matjaž, et al. "Evolutionary dynamics of group interactions on structured populations: a review." *Journal of the royal society interface* 10.80 (2013): 20120997.), the paper shines a new light on these observations from the perspective of a competition between local structure and global heterogeneity. This is a novel and very interesting contribution, in addition to the observation that heterogeneity can be detrimental to cooperation in certain settings.

Authors' response: We thank the referee for his/her time reviewing our manuscript, for the valuable feedback that has helped to substantially improve the manuscript, and for finding our work novel and considering it a very interesting contribution. In the following we provide detailed answers to all the points raised.

Is the work convincing, and if not, what further evidence would be required to strengthen the conclusions? On a more subjective note, do you feel that the paper will influence thinking in the field? Please feel free to raise any further questions and concerns about the paper. I think the paper makes an interesting point—that we find a trace of spatial selection in a new space derived from the network—and that in cases of strongly clustered structures, the heterogeneity of the network can hinder cooperation. What I think is still missing in the picture drawn from this work is an understanding (and possibly) quantification of what really is happening in the network and what are the main ingredients that control whether heterogeneity is good/bad for cooperation, what level of local cluster is required for cooperation to be stable etc. For example, the author notes that when the metric structure is strong the heterogeneity can be harmful to cooperation. Why does this happen? is it because the hubs existing behave as bridges between (hyperbolic)spatially separated regions hence promoting mixing and destabilising cooperator clusters? Is it because there are different dynamics happening at the interfaces between clusters? Or does that happen due to a different mechanism?

Authors' response: We thank the referee for these valuable comments. In the revised version of the manuscript, we have added a new Sec. IID where we provide an explanation for the survival of metric clusters, which we support with a quantitative analysis of the fraction of intercluster links.

The survival of metric clusters can be understood similarly to spatial selection in Euclidean space: cooperators survive in a cluster as it shields them from detecting neighbors. The fraction of intercluster links determines how well cooperators are shielded. This fraction depends on the size of the clusters as well as the mean local clustering and heterogeneity. For larger clusters, the relative surface area of the border in contact with adjacent defectors decreases, which shields them better and explains why larger clusters are more likely to survive (see new Fig. 5A in the revised manuscript). For a given size, two different mechanisms determine the fraction of intercluster links. Firstly, the greater the degree of heterogeneity, the larger the number of hubs. These nodes are connected to many other nodes, and therefore form long-range connections in the metric representation, which are likely to connect cooperators and defectors by bridging the border of metric clusters. In line with the intuition of the referee, this is the reason why more heterogeneity hinders the survival of metric clusters (Fig. 5B in the revised manuscript). Increasing the mean local clustering coefficient will reduce the temperature \bar{T} , which reduces the amount of long-range connections due to randomness, illustrated in the new Figs. 1A and B in the revised manuscript. Therefore, a higher degree of mean local clustering reduces the number of intercluster links, which in turn favors the survival of metric clusters as explained before (Fig. 5B in the revised manuscript).

Secondly, although I personally like very much the work related to the extraction of underlying spaces for networks, I wonder how much this abstraction can and will be received by the general community working in evolutionary game theory. This is due to a mixture of technical expertise required to understand the methods and of a certain lack of direct interpretation implicit in these embeddings. I imagine this will be a very popular paper with the stat-mech/network community, but I have a hard time gauging the potential impact in others.

Authors' response: We thank the referee for finding our paper interesting and expecting it to be popular within the stat-mech/network community. Regarding the doubts on the potential impact in other communities, we would like to point out that metric embeddings of complex networks have already been applied to game theory, see “Navigable networks as Nash equilibria of navigation games”, which has been published in Nature Communications [Nat. Com. 6, 7651] and has attracted a lot of attention (in the 97th percentile (ranked 17th) of the 662 tracked articles of a similar age in Nature Communications, as of today). Furthermore, papers building on these techniques have been published in journals with a very broad outreach, including Nature [Nat. 489, 537–540], Nature Physics [Nat. Phys. 5, 74 - 80 & Nat. Phys. 12, 1076–1081], and Nature Communications [Nat. Com. 8, 14103 & Nat. Com. 1, 62], in addition to the aforementioned article.

Finally, we believe that the additional analysis carried out in the revision providing an explanation for the findings and the illustrations in the new Figs. 1A and B in the revised manuscript will make it easier for the readers to develop an intuition regarding the metric space and its importance for the survival of cooperators.

**We would also be grateful if you could comment on the appropriateness and validity of any statistical analysis, as well the ability of a researcher to reproduce the work, given the level of detail provided.* I have one question/doubt regarding the measure of cluster timescale, which is not likely to change the final picture but I think it is important to raise. The author adopts a KS test between the angular distribution of cooperators/defectors and then weighs it with the cooperators fraction to extract a measure used to define the stability of the state. He mentions that the cooperator distribution is tested against a normal distribution, although I think he meant the uniform one (as per Eq. 8). Now my point is that I'm confused: if we use a normal distribution for the angular distribution of cooperators, I imagine that prior to mean that we are looking for a coop cluster that is localised in angular space; on the contrary, if we use the uniform distribution (which I deem more correct in this case) the measure is averaging over all the whole angular structure of the network, and measuring any large localisation of cooperators. However, this assumption implies that we consider the clustered state as the state of the whole network, rather than track the individual evolution of various spatial clusters (which instead really are the focus of the paper). Also, this type of measure has problems distinguishing configurations that have moving but fixed-density distributions of cooperators (e.g. a cluster that evolves rotating around the network would always have the same ρ , but it would be a very different cluster at each time). How much would it change if one adopted something along the lines of KL divergence between configurations at different times?*

Authors' response: The referee is right, we mean a uniform distribution, not a normal distribution. We apologize and have corrected this error in the revised manuscript.

We agree with the referee that the KL divergence is a more elegant measure to distinguish different distributions. Unfortunately, the KL divergence has problems if the histogram includes bins whose value is 0, which is often the case in the considered angular distribution of cooperators as they tend to form metric clusters.

We also agree that $\bar{\rho}$ only measures the global strength of the clustering, and that in principle there could be rotating clusters. We follow the suggestion of the referee and study the difference between configurations at different times (see new section in the Supplementary Materials). We find that when the system reaches the statistically stationary state, the clusters are at fixed positions, and the value of the KS-statistics between different timepoints is zero.

Regarding reproducibility, the author provided the data and pointers to the code needed to reproduce the paper (potentially on other network datasets), so the work is definitely reproducible by other researchers (although due to the nature of the embedding construction, it might not be very easy for researchers

outside of physics/math/compsci fields).

Authors' response: We thank the referee for the positive comment regarding our efforts to make our research reproducible.

**Overall assessment:* Based on the comments above, I consider this paper as a solid and interesting contribution. However, I harbour doubts about its potential impact to the broader community in consideration of the elements of interpretation that I highlighted above. So I would recommend a revision of the manuscript to address these concerns before reaching a final decision.*

Authors' response: We thank the referee for finding our work interesting. Regarding the potential impact to a broader community due to the use of metric embeddings of complex networks, please see our answer above pointing out previous publications in Nature Communications and similar journals that have used such methods and that have attracted a lot of attention from a broad readership as well as our efforts to make these findings more intuitive for a broad audience.

Finally, we again sincerely thank the referee for his/her extreme valuable comments that have helped to significantly improve the manuscript. To sum up, we have followed the suggestions of the referee and now provide a quantitative explanation for the survival of metric clusters (new Sec. IID in the revised manuscript), consider the difference of the angular distribution of cooperators between configurations at different times (new Sec. V in Supplementary Materials), have conducted a significantly more comprehensive analysis including different network and cluster sizes, and have made an additional effort to make the effect of metric spaces more intuitive for a broad audience (new Figs. 1A and B in the revised manuscript). We hope to have met his/her expectations with our answers and the comprehensive revision of the manuscript.

Referee C:

This paper studied the evolution of cooperation on social networks, which is one of the most popular topics in evolution game. Many theoretical studies have found that network heterogeneity favors cooperation. One of the main reasons behind the increase of cooperation in heterogeneous networks is that hubs are usually occupied by cooperators, which ensures their long-term success in the evolutionary process. In this paper, however, the author indicated that “spatial selection” can be even more efficient than heterogeneity in sustaining cooperation. Frankly speaking, I don’t understand why network heterogeneity is not a part of “spatial selection”. After reading the paper, I guess the author wants to say that if the local clustering coefficient in scale-free networks is large, then cooperators can survive better in local clusters than hubs.

As a general assessment, this paper has the potential of making a nice contribution to this field (i.e., how the local clustering coefficient affects cooperation). However, it has several problems. Specifically, I have two main comments, why clustering coefficient is important, and whether this result holds in general.

Authors’ response: We thank the referee for his/her time reviewing our manuscript and for considering that this paper has the potential of making a nice contribution to this field. We have significantly revised the manuscript addressing the points raised by the referee, and provide answers to each point in the following.

(1) In the discussion session, the author only summarizes the simulation outcomes, but did not discuss why and how the local clustering coefficient works. I think the author should provide a reasonable explanation for the main result. A possible choice would be to give some toy examples (see examples in Allen et al., 2017) to show that sometimes the hub cannot maintain cooperation but clusters can. This will help us to understand the role of local structure in promoting cooperation. Furthermore, a carefully discussion for the main result is necessary. In fact, it is counter intuitive that the hubs defect but cooperation can be stably maintained in leaves, because in this case the hubs will have higher payoffs than the leaves and the leaves will then imitate the hubs. Sanots et al., (2008) have provided a very good example (double-star graph) to demonstrate why the hub is crucial in promoting cooperation. So I think you should also give some examples to support your result.

Authors’ response: We thank the referee for these valuable suggestions. In the revised manuscript, we now explain why and how the local clustering coefficient works (Methods) and its relation to the metric space (Figs. 1A and B in the revised manuscript). We now provide a comprehensive explanation for the observed results (new Section IID in the revised manuscript), which we discuss and quantify in terms of the fraction of intercluster links.

Furthermore, we have followed the suggestion of the referee and now consider different toy examples (new Sec. VI in the Supplementary Materials). The considered examples include the mentioned double-star network as well as small and medium sized networks generated with the model described in the manuscript. These examples allow us to understand why and when cooperation is sustained, and we connect this intuition to an explicit realization of the system.

We sincerely thank the referee again for this valuable suggestion that has helped to improve the manuscript significantly.

(2) The second problem is whether this result holds in general. The author observed that under certain conditions, spatial selection becomes more important than heterogeneity. However, these observations are based on numerical simulations with some specific parameters or networks. Thus, the universality of this result is questionable. I do realize that it is very difficult to provide a rigorous mathematical analysis. But at least, the author should analyze some toy examples carefully, and derive the condition for these examples.

Authors’ response: In the revised version of the manuscript, we now conduct a significantly more comprehensive analysis, in particular including network sizes ranging from 10^3 to 10^5 . One key finding is that metric clusters survive only if they are large enough. Nevertheless, we have considered networks with power-law exponents $\gamma \in [2.1, 4]$, ranging from very heterogeneous and scale-free to less heterogeneous and not scale-free networks and with clustering coefficients in the range of $\bar{c} \in [0.3, 0.75]$. Finally, we take

into account the whole parameter range $T \in [0, 2]$ and $S \in [-1, 1]$ of the game parameters. We sincerely believe that our analysis can be considered very comprehensive.

Following the suggestions of the referee, in the revised manuscript, we now explain what leads to the survival of metric clusters (see new Sec. IID in the revised manuscript) and present an extensive analysis of toy examples (see new Sec. VI in the Supplementary Materials and the response to the previous point). We now also provide approximated conditions when metric clusters are likely to survive: clusters are larger than $\sim 10^3$ nodes, networks have $\bar{c} > 0.5$, and are not too heterogeneous ($\gamma > 2.5$).

Other issues:

1. The definition of “spatial selection” is confusing. From my point of view, “spatial selection” introduced by Nowak and May (1992) means that there is a spatial structure in the evolutionary process. The spatial structure can be homogenous or heterogeneous, and spatial clusters can be formed in both homogenous and heterogeneous networks. In particular, the hub can also form spatial clusters (e.g., double-star graph in Santos et al., 2008). Thus, I don’t understand why the author thinks spatial selection and network heterogeneity are two different things.

Authors’ response: We acknowledge that the term “spatial selection” might cause confusion. We have replaced it with “metric clusters” in the revised manuscript.

2. Section B is based on the stag hunt game with $T=0.5$ and $S=-0.5$. However, this game is not really a social dilemma. With these parameters, cooperation and defection is bistable, the mixed equilibrium is at $1/2$, and cooperation is payoff dominance. That is, cooperation should be better than defection. Thus, it is strange for me that the cooperation level is only 0.4 after 10^5 update steps. If you run the simulation for a longer time, I guess the cooperation level will finally converge to 1.

3. Continue to the above issue. It is better to start with the prisoner’s dilemma game in Section B. For me, Figure 4 is more interesting than Figure 2.

Authors’ response: We have followed the suggestions of the referee and replaced the stag hunt game with the prisoner’s dilemma.

Finally, we again thank the referee for his/her valuable feedback that has helped to significantly improve the manuscript. To sum up, we have followed the suggestions of the referee and now provide an extensive analysis of toy examples (new Sec. VI in the Supplementary Materials), present a quantitative explanation for the survival of metric clusters (new Sec. IID in the revised manuscript), have reframed the term “spatial selection”, and have conducted a significantly more comprehensive analysis including different network and cluster sizes. We hope to have met his/her expectations with our answers and the revision of the manuscript.

Reviewers' comments:

Reviewer #2 (Remarks to the Author):

In my opinion, the author has satisfactorily addressed the comments in the previous round of reviews. The revised manuscript presents a more substantial analysis and an enhanced explanation of the results and of the mechanisms behind them.

In consideration of this, I recommend the revised version of the manuscript for publication.

Reviewer #3 (Remarks to the Author):

Comments on "Metric clusters in evolutionary games on scale-free networks"

This paper has been greatly improved after the first revision, and all my comments have been addressed properly. I now understand why local clusters can maintain cooperation and why network heterogeneity sometimes hinders the evolution of cooperation.

I only have some small comments before the final acceptance.

(1) I don't understand Figure 2G. What is the meaning of x-axis? The frequency of cooperation, or sometime else? Furthermore, why they are only 7 data points for each color?

(2) In Figure 3 and related discussions, the authors chose 0.3 as a threshold value. For instance, the authors wrote "the degree of final cooperation exceeds an arbitrarily chosen threshold value of 0.3." However, I don't understand why 0.3 is a proper threshold. Is there any reason? From my point of view, 0.5 is better than 0.3. Since the initial cooperation level is 0.5, it makes sense to say that cooperation is maintained if the final level is not less than 0.5. From Figure 2 and Figure 4, the cooperation level can indeed reach 0.5 for networks with high local clustering. Thus, it would be better to consider 0.5 as the threshold value.

(3) In Figure S9 captions, please add the interpretation "Cooperators are marked by blue and defectors by red."

September 20, 2017

Revised manuscript “Metric clusters in evolutionary games on scale-free networks”**Response to Referees’ Comments****Referee #2:**

In my opinion, the author has satisfactorily addressed the comments in the previous round of reviews. The revised manuscript presents a more substantial analysis and an enhanced explanation of the results and of the mechanisms behind them. In consideration of this, I recommend the revised version of the manuscript for publication.

Authors’ response: We thank the referee for his/her positive opinion and for recommending our manuscript for publication.

Referee #3:

This paper has been greatly improved after the first revision, and all my comments have been addressed properly. I now understand why local clusters can maintain cooperation and why network heterogeneity sometimes hinders the evolution of cooperation. I only have some small comments before the final acceptance.

Authors’ response: We thank the referee for the very positive comment. In the following we provide detailed answers to the small comments mentioned by the referee. We would also like to point out that we have added an explanation of the difference between metric clusters and connected clusters using an explicit example (Fig. 3 in the revised manuscript and text marked by blue, as well as Supplementary Fig. S3 and related text).

1) I don’t understand Figure 2G. What is the meaning of x-axis? The frequency of cooperation, or sometime else? Furthermore, why they are only 7 data points for each color?

Authors’ response: We agree with the referee that this point was not explained clearly in the manuscript. The plot comprises several histograms. Each color represents the histogram of cooperation among different realizations of the system at time t , where the color denotes the time (see legend in plot H). The 7 points hence represent 7 bins. The x -axis denotes the cooperation density C and the y -axis shows the probability $P(C)$ that in one realization of the system the density of cooperation at time t is C . We have added this explanation to the caption of the figure.

2) In Figure 3 and related discussions, the authors chose 0.3 as a threshold value. For instance, the authors wrote “the degree of final cooperation exceeds an arbitrarily chosen threshold value of 0.3.” However, I don’t understand why 0.3 is a proper threshold. Is there any reason? From my point of view, 0.5 is better than 0.3. Since the initial cooperation level is 0.5, it makes sense to say that cooperation is maintained if the final level is not less than 0.5. From Figure 2 and Figure 4, the cooperation level can indeed reach 0.5 for networks with high local clustering. Thus, it would be better to consider 0.5 as the threshold value.

Authors’ response: The reason for choosing 0.3 as a threshold in Fig. 3 (Fig. 4 in the revised

manuscript) is to make it better comparable with the results in Fig. 4 (Fig. 5 in the revised manuscript). There, we have chosen 0.3 because we want to distinguish where metric clusters lead to a behavior that is qualitatively different as in the case of random or hub preferential initial conditions. For finite system size, cooperation can be sustained although the mean final cooperation density is smaller than 0.5, which occurs for example for the parameters shown in Fig. 4D (Fig. 5D in the revised manuscript). We believe that 0.3 is a good trade-off in the sense that it is significantly larger than 0 (cooperation is sustained, or cooperators survive), and at the same time it identifies with sufficient precision the area in the parameter space where metric clusters show a qualitatively different behavior as compared to the other initial conditions. Finally, note that the only purpose of the threshold value is to present the results in a more compact way, and the exact values of the mean final cooperation are shown in Figs. 4A-C (Fig. 5A-C in the revised manuscript) and in Supplementary Fig. S2.

3) In Figure S9 captions, please add the interpretation "Cooperators are marked by blue and defectors by red."

Authors' response: We thank the referee for pointing this out and we have added the sentence to the caption of the figure, which is Supplementary Fig. S10 in the revised Supplementary Materials.

Finally, we want to thank the referee again for his/her positive opinion about our work and especially for the very constructive suggestions in the last round of reviews that have helped to improve the manuscript substantially. We hope to have met his/her expectations with our answers to the small comments and that he/she feels that the manuscript is now ready for publication.

Reviewers' Comments:

Reviewer #1:

None

Reviewer #3:

Remarks to the Author:

all my comments have been addressed, ready for publication

Revised manuscript "Metric clusters in evolutionary games on scale-free networks" Response to Referees' Comments Referee C: all my comments have been addressed, ready for publication
Authors' response: We thank the referee again for his/her time reviewing our manuscript and for considering it ready for publication.